# Monitoring physiological processes of fast-growing broilers during the whole life cycle: Changes of redox-homeostasis effected to trassulfuration pathway predicting the development of non-alcoholic fatty liver disease

**Georgina Pesti-Asbóth**[1], **Endre Szilágyi**[1], **Piroska Bíróné Molnár**[1], **János Oláh**[2], **László Babinszky**[3], **Levente Czeglédi**[4], **Zoltán Cziáky**[5], **Melinda Paholcsek**[1], **László Stündl**[1], **Judit Remenyik**[1]*

**1** Faculty of Agricultural and Food Sciences and Environmental Management, Institute of Food Technology, University of Debrecen, Debrecen, Hungary, **2** Farm and Regional Research Institute of Debrecen, University of Debrecen, Debrecen, Hungary, **3** Faculty of Agricultural and Food Sciences and Environmental Management, Department of Animal Nutrition Physiology, Institute of Animal Science, Biotechnology and Nature Conservation, University of Debrecen, Debrecen, Hungary, **4** Faculty of Agricultural and Food Sciences and Environmental Management, Department of Animal Science, Institute of Animal Science, Biotechnology and Nature Conservation, University of Debrecen, Debrecen, Hungary, **5** Agricultural and Molecular Research and Service Group, University of Nyíregyháza; Nyíregyháza, Hungary

* remenyik@agr.unideb.hu

## Abstract

In the broiler industry, the average daily gain and feed conversion ratio are extremely favorable, but the birds are beginning to approach the maximum of their genetic capacity. However, as a consequence of strong genetic selection, the occurrence of certain metabolic diseases, such as myopathies, ascites, sudden cardiac death and tibial dyschondroplasia, is increasing. These metabolic diseases can greatly affect the health status and welfare of birds, as well as the quality of meat. The main goal of this study was to investigate the changes in the main parameters of redox homeostasis during the rearing (1–42 days of age) of broilers with high genetic capacity, such as the concentrations of malondialdehyde, vitamin C, vitamin E, and reduced glutathione, the activities of glutathione peroxidase and glutathione reductase, and the inhibition rate of superoxide dismutase. Damage to the transsulfuration pathway during growth and the reason for changes in the level of homocysteine were investigated. Further, the parameters that can characterize the biochemical changes occurring in the birds were examined. Our study is the first characterize plasma albumin saturation. A method was developed to measure the levels of other small molecule thiol components of plasma. Changes in redox homeostasis induce increases in the concentrations of tumor necrosis factor alpha and inflammatory interleukins interleukin 2, interleukin 6 and interleukin 8 in broilers reared according to current large-scale husbandry technology and feeding protocols. A significant difference in all parameters tested was observed on the 21st day. The concentrations of cytokines and homocysteine increased,

**Data Availability Statement:** All relevant data are within the paper and its Supporting Information files.

**Funding:** This study was financially supported by the Gazdaságfejlesztési és Innovációs Operatív Program (GINOP) GINOP-2.3.2-15-2016-00042 project of the Széchenyi 2020 Program given by the European Union and the Hungarian Government. The funders had no role in study design, data collection and analysis, decision to publish, or preparation of the manuscript.

**Competing interests:** The authors have declared that no competing interests exist.

while the concentrations of glutathione and cysteine in the plasma decreased. Our findings suggest that observed changes in the abovementioned biochemical indices have a negative effect on poultry health.

## Introduction

The main issues that burden the broiler industry today might be caused by genetic selection strategies, such as a rapid growth rate, a decreasing slaughter age and facilitating feed efficiency [1]. Indeed, achieving extreme growth potential is a serious biotic stress factor that is accompanied by health problems leading to systemic diseases [2,3].

The path mechanisms of most common infectious diseases, including pneumonia, sepsis and enteritis, are directly linked to the failure of redox homeostasis [4–7]. According to the latest research findings, a high susceptibility to pathogens was closely related to oxidative stress in 308 Ross broiler chickens [8,9]. In addition, fatal complications, such as acidosis, sudden cardiac death, dyschondroplasia or wooden breast (WB) syndrome, have been proven to be a result of genetics along with additional unknown causes [10,11]. Therefore, revealing potential tools to avoid the accumulation of genetic alterations has recently become crucial.

In addition to being an important economic factor, the present condition of the poultry industry has drawn attention to global issues that must be solved, such as animal welfare. For instance, limiting the use of antibiotics seems to be challenging because genetically modified poultry strains are highly susceptible to infections [12].

Pathological alterations in the cardiovascular system and the liver are as important as infections in terms of the sustainability of the poultry industry. WB myopathy is a systemic disease in which complex metabolic and physiological changes have been implicated that is characterized by hardened areas, white striping and small hemorrhages in the pectoralis major muscle [13]. Macroscopic changes might develop due to the abnormal accumulation of endomysial and perimysial connective tissue, which in turn can lead to fibrosis, hypoxia, oxidative stress and the activation of inflammatory pathways [13–16]. Xing et al. [17] noted that WB syndrome is strongly correlated with hepatocyte damage and elevated serum levels of aspartate aminotransferase (AST) and γ-glutamyl transferase (GGT) accompanied by a high systemic inflammatory cytokine load. In addition, a high concentration of homocysteine (Hcy) was detected in animals suffering from myopathy compared to healthy chickens [18].

Other multifactorial diseases that affect production traits are ascites (AS) and sudden death syndrome, which are highly correlated with each other [19]. Cardiac arrest could also occur owing to hyperlipidemia-mediated atherosclerosis [20,21]. These conditions are often accompanied by liver fibrosis, which was originally common among birds bred in high-altitude areas. Currently, liver fibrosis has also become widespread in industrial farming, probably because of a deficiency in oxygen consumption that has been linked to the fast growth rate [22]. Severe hepatic impairment was reported in broilers with cold-induced ascites [23]. In addition, chronic inflammatory events have resulted in metabolic imbalance, as indicated by observed threefold elevation in serum Hcy [24].

During broiler growth, it is important to monitor how the antioxidant system can be characterized during each development cycle and how it correlates with the events in the transsulfuration pathway. To draw a more complete picture, it may be important to examine new parameters influencing the redox system. Recent research in the human body has shown that the small molecule thiol compounds in albumin and plasma significantly, if not mostly,

determine plasma redox homeostasis [25,26]. Although it may seem that this is not of great importance from an economic point of view, it is possible to prevent the phenotypic appearance of certain diseases by monitoring changes in biochemical parameters.

Worldwide, remediation of the untenable circumstances in the broiler industry has already started in the fields of health care management, environmental inspections and animal welfare accreditation systems, but the key points in bird growth where the prevention or reversal of malformations is feasible are still not clear [19,27,28]. Proper livestock farming technology is essential for minimizing abiotic stress factors, but limiting biotic stress is also important [29], and the latter is unattainable without revealing the mechanisms behind the development of genetic diseases.

In fact, novel biomarkers need to be found to facilitate the development of the poultry sector. This might also pave the way for the application of herbal substances as effective compounds that can modulate above mentioned molecular targets and in turn promote antibiotic-free livestock care when considering animal welfare and health protection. Therefore, the aim of the present study was to determine how the antioxidant status of fast-growing broilers changes during their lifecycle and what kind of enzymes (glutathione peroxidase and glutathione reductase) are activated to maintain homeostasis.

A further aim was to study how the transsulfuration pathway (TSP) is connected with the antioxidant protection system and its relevance in diagnostics.

## Materials and methods

### Ethical approval

The study was conducted with the approval of the local ethics committee of the University of Debrecen (DEMAB/12-7/2015).

### Birds, bird housing, experimental design, and diet

A total of 180 day-old ROSS 308 hybrid chickens of both sexes from a commercial hatchery in Hungary were used in this study. The experiments were carried out on the experimental farm of the University of Debrecen. All broilers were placed in the same barn. Chickens were kept in floor pens covered with wood shavings in a thermostatically controlled house at a stocking density of 650 cm$^2$/bird. The temperature was 32˚C at placement and gradually decreased by 1.5˚C/week. The birds were exposed to light based on Ross Broiler Management Handbook (2018) as follows: 23 L:1D during the first 7 days, 20 L:4D between Days 8–28 and a prolonged third phase such as 23 L:1D between Days 29–42 (L = light, D = dark) [30].

The day old chickens were placed into 3 pens (60 birds/pen). The experiment started at 1 day of age and lasted until 42 days of age. Broilers were fed a commercial maize–soybean-based basal diet (BD) free of antibiotics according to four feeding periods: prestarter (1 to 9 days), starter (10 to 21 days), grower (22 to 31 days), and finisher (32 to 42 days). All diets were fed in mash form. The composition and energy and nutrient contents of the diets are given in Table 1.

Diets and water were available *ad libitum* during the entire experiment. Broilers were weighed at 1, 10, 21, 32, and 42 days of age at which times the growth performance parameters, average body weight (BW) (6 birds/pen), average daily gain (ADG) (6 birds/pen), average daily feed intake (ADFI) and feed conversion ratio (FCR) were determined. The BW and ADG were measured individually, while the ADFI and FCR were calculated by pen at 1, 10, 21, 32 and 42 days of age.

**Table 1. Ingredients and chemical compositions of the basal diets fed during the prestarting (1–9 days), starting (10–21 days), growing (22–35 days) and finishing (35–42 days) phases.**

| Ingredient (%) | Prestarter (Days 1–9) | Starter (Days 10–21) | Grower (Days 10–21) | Finisher (Days 32–42) |
|---|---|---|---|---|
| Corn | 33 | 34 | 33 | 32 |
| Wheat | 27 | 29 | 31 | 32 |
| Soybean meal, solvent extracted (46.0% CP[a]) | 29 | 24 | 20 | 16 |
| Soybean meal, extruded (46.0%CP) | 4 | 6 | 4 | 4 |
| Sunflower meal, extracted | - | 1 | 3 | 4 |
| Feed yeast | 1 | - | - | - |
| Distillers-dried grains with solubles (DDGS) | - | 1 | 3 | 5 |
| Plant fats | 2 | 1 | 3 | 4 |
| Premix | 4 | 4 | 3 | 3 |
| Total | 100 | 100 | 100 | 100 |
| **Energy and nutrients in the diet** | | | | |
| Dry matter, % | 89.06 | 89.03 | 89.15 | 89.15 |
| $AME_n$ [b], Poultry, MJ/kg | 12.23 | 12.47 | 12.81 | 13.01 |
| Crude protein, % | 21.58 | 20.28 | 19.05 | 18.28 |
| Crude fat, % | 4.61 | 4.83 | 6.22 | 6.83 |
| Crude fiber, % | 3.37 | 3.51 | 3.7 | 3.88 |
| Lysine, % | 1.37 | 1.27 | 1.17 | 1.09 |
| Methionine, % | 0.57 | 0.54 | 0.53 | 0.49 |
| Methionine + cysteine, % | 0.94 | 0.9 | 0.87 | 0.83 |
| Calcium, % | 0.85 | 0.73 | 0.71 | 0.67 |
| Phosphorus, % | 0.63 | 0.55 | 0.52 | 0.49 |
| Sodium, % | 0.17 | 0.16 | 0.16 | 0.16 |
| Sodiumchloride, % | 0.282 | 0.252 | 0.242 | 0.244 |
| Vitamin A, mg/kg | 12.500 | 12.500 | 12.500 | 8.750 |
| Vitamin E, mg/kg | 50.001 | 50.001 | 50.001 | 35 |
| Vitamin $D_3$, mg/kg | 3.000 | 3.000 | 3.000 | 2.100 |

CP, crude protein; $AME_n$, apparent metabolizable energy corrected for zero nitrogen balance.

The animals were sacrificed by cervical dislocation. Blood and liver samples were collected from single bird per pen except for on Day 3, when three birds/pen were processed. Blood samples from Day 3 were pooled for the experiments.

The blood samples from Day 3 were pooled for the experiments. Blood was collected into EDTA-coated vacutainer tubes (BD, Franklin Lakes, NJ.). One milliliter of each sample was centrifuged at $1000 \times g$ and $4°C$ for 10 min. The supernatant plasma was further divided into 300 μL aliquots and stored at -80°C until analysis. Samples were used to determine the antioxidant status of the plasma.

Further, liver tissues were preserved in formalin (10% reagent grade for histology, neutral-buffered; VWR Hungary).

## Determination of antioxidant parameters

**Lipid peroxidation.** Lipid peroxidation was determined using a commercially available assay kit (ab118970, Abcam, Cambridge, United Kingdom) [31]. The measurements were based on the reaction between MDA and thiobarbituric acid (TBA) [32]. All reagents and

standard solutions were prepared according to the manufacturer's instructions. Plasma samples (20 μl) were mixed with 500 μL of 42 mM $H_2SO_4$, 125 μL of phosphotungstic acid solution was added, and each sample was mixed by vortexing and incubated at room temperature for 5 minutes. After incubation, we centrifuged the samples at $13000 \times g$ for 3 minutes. The pellet was collected and resuspended in 100 μL of distilled water on ice with 2 μL of butylated hydroxytoluene (BHT) (100-fold dilution). The final volume was adjusted to 200 μL with distilled water. After sample preparation, the assay was performed. The MDA-TBA adduct was generated in each sample and standard solution. The TBA reagents (600 μl) were added to the samples and standards for incubation at 95°C for 60 minutes. The reaction mixes were cooled in an ice bath for 10 minutes after incubation. The samples and standards were placed in duplicate in a 96-well microplate for analysis. The absorbance of the MDA-TBA adduct was measured at 532 nm, and the calculated concentrations are expressed in nmol/ml.

**Vitamin E.** A competitive enzyme immunoassay technique was used to determine the plasma vitamin E concentration [33] (E12V0032, BlueGene Biotech., Shanghai, China). The reagents were prepared as described in the manual. First, 100 μl of plasma, standard and blank were added to a microtiter plate, each in duplicate. The plate was incubated for 90 minutes at 37°C. After incubation, a washing step was performed two times. Then, 100 μl of vitamin E antibody was added to each well, and the plate was incubated again for 60 minutes at 37°C. After incubation, the contents of the wells were washed three times, and enzyme conjugate (100 μl) was added to each well. The plate was covered and incubated for 30 minutes at 37°C. Then, the contents of the ELISA plate were washed five times, and 100 μl of color reagent liquid was added to each well. The plate was incubated for 12 minutes (the protocol mentioned that the incubation time should be controlled within 30 minutes) at 37°C, and 100 μl of color reagent C was pipetted into each well. The absorbance values were measured at 450 nm, and the vitamin E concentrations are expressed in μg/ml.

**Vitamin C.** To determine the vitamin C concentration, a plasma-specific assay kit was used (ab65656, Abcam, Cambridge, United Kingdom) [34–36]. The measurements were performed according to the protocol in the manual with some modifications. The kit's supernatant assay buffer was set to pH 7.0 with NaOH (10 M). The kit provides a sensitive method for vitamin C measurement. Plasma vitamin C reduces $Fe^{3+}$ to $Fe^{2+}$, which shows strong absorbance that can be monitored between 545–600 nm. The reagent and standard were prepared as described in the protocol. Fifty microliters of each plasma sample, run in duplicate, was diluted two fold for measurement. The assay procedure was followed as described in the manual. The absorbance of each sample was measured at 593 nm, and the concentrations were calculated as recommended and are expressed in nmol/ml.

**Reduced glutathione.** The GSH concentration was measured by the enzymatic recycling method using a specific assay kit(703002,Cayman Chemical Company, Ann Arbor, MI, United States) [37–39].This assay is based on the reaction between5,5'-dithio-bis (2-nitrobenzoic acid) (DTNB) and GSH and generates a yellow product (2-nitro-5-thiobenzoic acid). All of the reagents and standards were prepared as described in the "reagent preparation" and "standard preparation" sections. The assay was performed following the protocol in the manual. We used 50 μl plasma samples for the measurements. The concentration of GSH was determined by measuring the absorbance at 405 nm. The concentration of GSH in each sample was determined by comparison with a GSH standard curve and is expressed in μM.

**Glutathione peroxidase.** The activity of GPx was measured with a commercially available kit(ab102530, Abcam, Cambridge, United Kingdom) [40–44]. In the assay, GSSG produced after GPx oxidizes GSH during $H_2O_2$ reduction. GSSG is reduced back to GSH by GR with the aid of nicotinamide adenine dinucleotide phosphate (NADPH). The reduction of NADPH is proportional to GPx activity and can be measured colorimetrically at 340 nm. The kit reagents

were dissolved as described in the manual. A standard curve was prepared as described in the kit [45]. Measurements were performed according to a previous study [46]. The activity of GPx is expressed as mU/ml.

**Glutathione reductase.** The activity of GR in plasma was determined using a specific assay kit(ab83461, Abcam, Cambridge, United Kingdom) [47]. In this assay, GSH is formed from GSSG by GR; then, GSH reacts with DTNB, and a 2-nitro-5-thiobenzoate anion (TNB$^{2-}$) is generated [45]. The change in absorbance was measured at 405 nm. The kit reagents were dissolved as described in the "components and storage" section. The protocol recommends pretreating the samples. First, 5 μL of 3% $H_2O_2$ was added to 100 μL of each sample. The samples were incubated at 25˚C for 5 min. Then, 5 μL of catalase was added to each sample, and we incubated the samples again at 25˚C for another 5 min. After the pretreatment procedure, 50 μL of each pretreated sample was added to the sample wells. The standard curve was prepared as described in the manual, and 50 μL of diluted standard solution was added to each well. Measurements were performed according to a previous study [46]. GR activity is expressed in nmol/min/mL = mU/ml.

**SOD inhibition rate.** The superoxide dismutase(SOD)inhibition rate was measured with a specific assay kit(ab65354, Abcam, Cambridge, United Kingdom) [48–51]. In this assay, xanthine oxidase produced superoxide anions, and the conversion of superoxide anions into hydrogen peroxide was catalyzed by SOD. Superoxide anions and the water-soluble tetrazolium salt WST-1 can react to produce a water-soluble formazan dye, the absorbance of which was detected at 450 nm. The reagents were prepared as described in the kit. First, 20 μL of Blank 1, Blank 2, Blank 3 or sample was added to a 96-well plate in duplicate. Then, WST-1 solution (200 μl) was added to each blank and sample. Twenty microliters of dilution buffer were added to the Blank 2 and Blank 3 solutions. Enzyme working solution (20 μl) was added to each sample and Blank. Then,the plate was incubated at 37˚C for 20 minutes. The absorbance (A) was then measured at 450 nm. We used the following equation to calculate the SOD inhibition rate:

$$SOD\ inthibitation\ rate\ (\%) = \frac{(A_{blank\ 1} - A_{blank\ 3}) - (A_{sample} - A_{blank\ 2})}{(A_{blank\ 1} - A_{blank\ 3})} *100$$

**Free thiol concentration.** The free thiol concentration was determined through Ellman's assay (Ethos Biosciences Inc., Philadelphia, PA, United States). The kit is based on the reaction between DTNB and disulfide bonds to produce 2-nitro-5-thiobenzoic acid, and the absorbance of this product can be measured at 405 nm [52–55]. The reagents and standard were prepared as recommended in the manual. Fifty microliters of each sample (15-fold dilution) or standard was added into the appropriate well in duplicate. Then, 150 μl of Ellman's reagent was pipetted into each well. The absorbance was measured immediately at 405 nm. The concentration of free thiols in each sample was determined by comparing the absorbance values of the samples to a standard curve. The free thiol concentration is expressed as the DTNB-thiol concentration in μM.

**A**lbumin. The albumin concentration was determined using a commercially available reagent (4125S, Diagnosticum Zrt., Budapest, Hungary). The protocol was based on the method of Doumas et al. (1971) [56]. Plasma samples (3 μl/sample) and working reagent (300 μl) were mixed and incubated in a microplate at 37˚C for 3 min. Then, the absorbance was measured at 628 nm. The following equation was used to calculate the albumin concentration:

$$Albumin\ (g/l) = \frac{A_{sample}}{A_{standard}} *concentration\ of\ the\ standard$$

The abovementioned parameters were measured colorimetrically with a spectrophotometer (Spentrophotostar$^{nano}$, BMG Labtech). The results were calculated using MARS data analysis software.

## Determination of aspartate aminotransferase (AST) levels

Plasma samples (200 μl) were pipetted into Hitachi cups (10394246001) and assessed using ASTL reagent (20764949322, ASTL 500T Cobas C Integra) [57]. Both the cup and ASTL reagent were from ROCHE Ltd. (Budapest, Hungary). The AST level was determined by measuring the decrease in absorbance using Cobas C 311 instrument (ROCHE Ltd., Budapest, Hungary).

## Determination of total thiol and disulfide contents

Xiaoyun Fu et al. [58] developed a new method for thiol and disulfide content determination, which was used here. To avoid thiol-disulfide exchange and artefactual oxidation thiols were blocked with NEM. The NEM-treated samples were reduced 12 mM 1,4-Dithiothretiol (DTT) (in 20 mM phosphate buffer, pH 7.4) and incubated for 20 min at 65°C.Then 60 mM DTT (5 mM phosphate buffer, pH 6.5) were added and the samples were incubated again for 30 min at 37°C.The proteins were precipitated with 80% v/v methanol for an hour at -20°C. The samples were centrifuged and the supernatants were used for the LC-MS analysis. Samples were prepared for cysteine (Cys), homocysteine (Hcy), cysteinyl-glycine (Cys-Gly), γ-glutamylcysteine (γ-GC), cystine (CySS), glutathione (GSH) and N-acetylcysteine (NAC) analysis.

**Preparation of standard solutions.** A standard mixture (10 μg/ml for each compound) was prepared from the Cys, Hcy, Cys-Gly, γ-GC, CySS, and GSH standards (168149-25g, 69453-10mg, C01666-25mg, G0903-25mg, C8630-1g, and PHR1359-500mg; Merck Life Science Ltd., Budapest, Hungary). NAC (A15409.14) was purchased from VWR International Ltd., Debrecen, Hungary.

To 150 μl of standard mixture were added 300 μl of N-ethylmaleimide (NEM) solution (100 μg/ml), 1020 μl of water and 30 μl of formic acid solution (0.01% V/V). The reaction mixture was incubatedat 37°C for 30 minutes. After cooling, standard solutions were prepared at seven concentrations (0.1, 1, 5, 10, 25, 50 and 100 ng/ml) by diluting with water.

## LC–MS analysis

Analyses were performed using a Dionex Ultimate 3000RS UHPLC system (Thermo Fisher, Waltham, MA, USA) coupled to a Thermo Q Exactive Orbitrap hybrid mass spectrometer equipped with an Acclaim Mixed-Mode HILIC-1analytical column (2.1 × 150 mm, 3 μm particle size). The flow rate was maintained at 0.3 mL/min. The column oven and post column cooler temperatures were set to 25°C ± 1°C. The temperature of the samples was 25°C ± 1°C. The mobile phase consisted of water (A) and methanol (B) (both acidified with 0.1% formic acid). The gradient elution program was as follows: 0–1 min, 95% A; 1–6 min, →0% A; 6–10 min, 0% A; 10–10.5 min, →95% A; 10.5–20 min, 95% A. The injection volume was 5 μL.

The Thermo Q Exactive Orbitrap hybrid mass spectrometer (Thermo Fisher, Waltham, MA, USA) was equipped with a HESI source. The samples were analyzed in positive ion mode using the selected ion monitoring (SIM) technique with the following inclusion list ([M + H]$^+$): cysteine-NEM ($C_9H_{15}N_2O_4S$): 247.07525; N-acetylcysteine-NEM ($C_{11}H_{17}N_2O_5S$): 289.08582; homocysteine-NEM ($C_{11}H_{17}N_2O_5S$): 261.09090; cysteinylglycine-NEM ($C_{11}H_{18}N_3O_5S$): 304.09672; γ-glutamylcysteine-NEM ($C_{14}H_{22}N_3O_7S$): 376.11785; cystine ($C_6H_{13}N_2O_4S_2$): 241.03167; glutathione-NEM ($C_{16}H_{25}N_4O_8S$): 433.13931; and glutathione disulfide ($C_{20}H_{33}N_6O_{12}S_2$): 613.15979.

The capillary temperature of the mass spectrometer was set to 320˚C, the spray voltage was 4.0 kV and the resolution of 35,000. The sheath gas flow rate and aux gas flow rate were 32 AU and 7 AU, respectively. The differences between the measured and calculated monoisotopic molecular masses were less than 5 ppm in each case. The data were acquired and processed using Thermo Xcalibur 4.0 software (ThermoFisher, Waltham, MA, USA) [58].

## Determination of cytokine concentrations

The concentration of interleukin 2 (IL-2) was determined using a commercially available sandwich ELISA kit (E0003Ch, Bioassay Technology Laboratory, Junjiang International Bldg., Shanghai China). In this ELISA, the plate was pre coated with chicken IL-2 antibody, and the sample was added so that the IL-2 in the sample could bind to the antibodies coated on the wells. The reagents were prepared as described in the manual. Fifty microliters of standard solutions and 40 μl of sample were added into the appropriate wells. IL-2 antibody (10 μl) was pipetted into the sample wells, and 50 μl of streptavidin-HRP was added to the sample wells and standard wells (but not the blank control wells). The plate was covered and incubated for 60 minutes at 37˚C. After incubation, the cover was removed, and the contents of the plate were washed 5 times with wash buffer (350 μl); each washing step took 1 minute. Fifty microliters of substrate solution A and 50 μl of substrate B were added to each well. The plate was sealed and incubated for 10 minutes at 37˚C in the dark. Then, stop solution (50 μl) was pipetted into each well, and the blue color immediately changed to yellow. The optical density (OD value) of each well was determined immediately at 450 nm using a microplate reader (within 10 minutes after adding the stop solution). The free IL-2 concentration is reported as ng/l.

The levels of interleukin 6 (IL-6), interleukin 8 (IL-8) and tumor necrosis factor-α were determined using commercially available sandwich ELISA kits (E0004Ch, E0005Ch, and E0025Ch) from Bioassay Technology Laboratory, Junjiang International Bldg.,Shanghai China. All of the ELISA kit principles and assay procedures were the same as those of the IL-2 ELISA kit [59–62].

## Histological examinations

Formalin-fixed liver tissue samples were embedded in paraffin. Four-micron-thick sections were generated with a rotary microtome and routinely stained with hematoxylin and eosin [63].

## Statistical analysis

The normality assumption was checked by analyzing the ANOVA model residuals using Shapiro-Wilk test. The homogeneity of variance was checked using Levente's test. Statistical analyses and data visualization were performed using GraphPad Prism (version 9). The growth performance parameters, antioxidant parameters and concentrations of total thiols, disulfides and cytokines were analyzed by one-way analysis of variance (ANOVA) followed by Tukey's method for multiple comparisons. A value of $p < 0.05$ was considered to indicate statistical significance, and the results are presented as the mean ± SEM. The correlation between biochemical parameters was analyzed by Pearson's correlation ($p < 0.05$).

## Results and discussion

### Growth performance

During the four phases of feeding, the body weight of the birds was 232 g on the 9th day, which tripled on Day 21 (759g, $p < 0.0001$). A further significant increase in weight occurred

**Table 2. Growth performance of the broiler chickens.**

| | Prestarter (Days 1–9) | | Starter (Days 10–21) | | Grower (Days 22–31) | | Finisheru (Days 32–42) | | p-value |
|---|---|---|---|---|---|---|---|---|---|
| | Mean[a] | SEM | Mean | SEM | Mean | SEM | Mean | SEM | |
| BW (g) | 232 | 3,32 | 759*** | 10.87 | 1713 | 24.54 | 275***8 | 39.51 | <0.001 |
| ADG (g/day/bird) | 19 | 0.27 | 48*** | 0.75 | 87*** | 1.25 | 104*** | 1.49 | <0.001 |
| ADFI (g/day/bird) | 4 | 0.06 | 50*** | 0.72 | 130*** | 1.86 | 114*** | 1.63 | <0.001 |
| FCR (kg feed/kg BW) | 0.19 | 0.003 | 1.35*** | 0.019 | 1.23*** | 0.018 | 1.26*** | 0.018 | <0.001 |

[a]Mean values. BW and ADG are based on individual values (n = 18).The ADFI was calculated per pen (n = 3)

*P<0.05

** p<0.005

*** p<0.001.

during the 3rd stage of growth. By the end of the experiment, the average weight of the birds was 2758 g ($p<0.0001$) (. Dietary indicators related to growth performance, such as body weight (BW), average daily gain (ADG), average daily feed intake (ADFI) and feed conversion ratio (FCR), were measured (Table 2). Significant differences were observed by comparing the data from each time point with the data from the first time point for all parameters [64,65].

The mortality of the birds was monitored and found to be very low (0.56%), and there was no association between mortality and nutrition. No veterinary treatment was required for the entire duration of the experiment.

The results of the current study indicate that the performance of the experimental subjects corresponds to the values typical of fast-growing broilers. The data also show that the performance of broilers did not indicate the presence of metabolic diseases among the livestock [66–69].

## Redox parameters

Genetic selection of broilers facilitates animal susceptibility to biotic and abiotic stress factors. Oxidative stress is known to be one of the most important stressors and can be accompanied by serious outcomes; thus, it is among the most studied parameters [1,2]. Oxidized thiol compounds can be considered reactive oxygen species (ROS) that affect cellular calcium ($Ca^{2+}$) homeostasis, which in turn leads to structural damage to $Ca^{2+}$ ATPase. Accumulation of $Ca^{2+}$ can cause damage to the endoplasmic reticulum (ER), which might induce multiple pathological alterations [70,71].

The impact of oxidative stress on physiological functions has received exceptional scientific interest for decades. Its characteristic parameters, namely, (i) enzyme activities (SOD, catalase, glutathione-S-transferase, GR and GPx), (ii) concentrations of small antioxidant molecules (vitamins E and C and GSH), and (iii) the expression of heat shock proteins, enable us to study the role of a specific substance or environmental factor in the maintenance of redox homeostasis [72,73]. However, the dynamic characteristics of redox homeostasis should be taken into consideration, meaning that accurate conclusions cannot be drawn regarding the oxidant/prooxidant balance and pathological conditions from measurements taken at only one-time point. The status of redox homeostasis must be followed during the whole life cycle, which makes it possible to compare different developmental stages.

In samples taken at the first time point, the plasma MDA concentration was 187 nmol/ml (Fig 1A). In the plasma of 21-day-old animals, this value had decreased, but then it increased

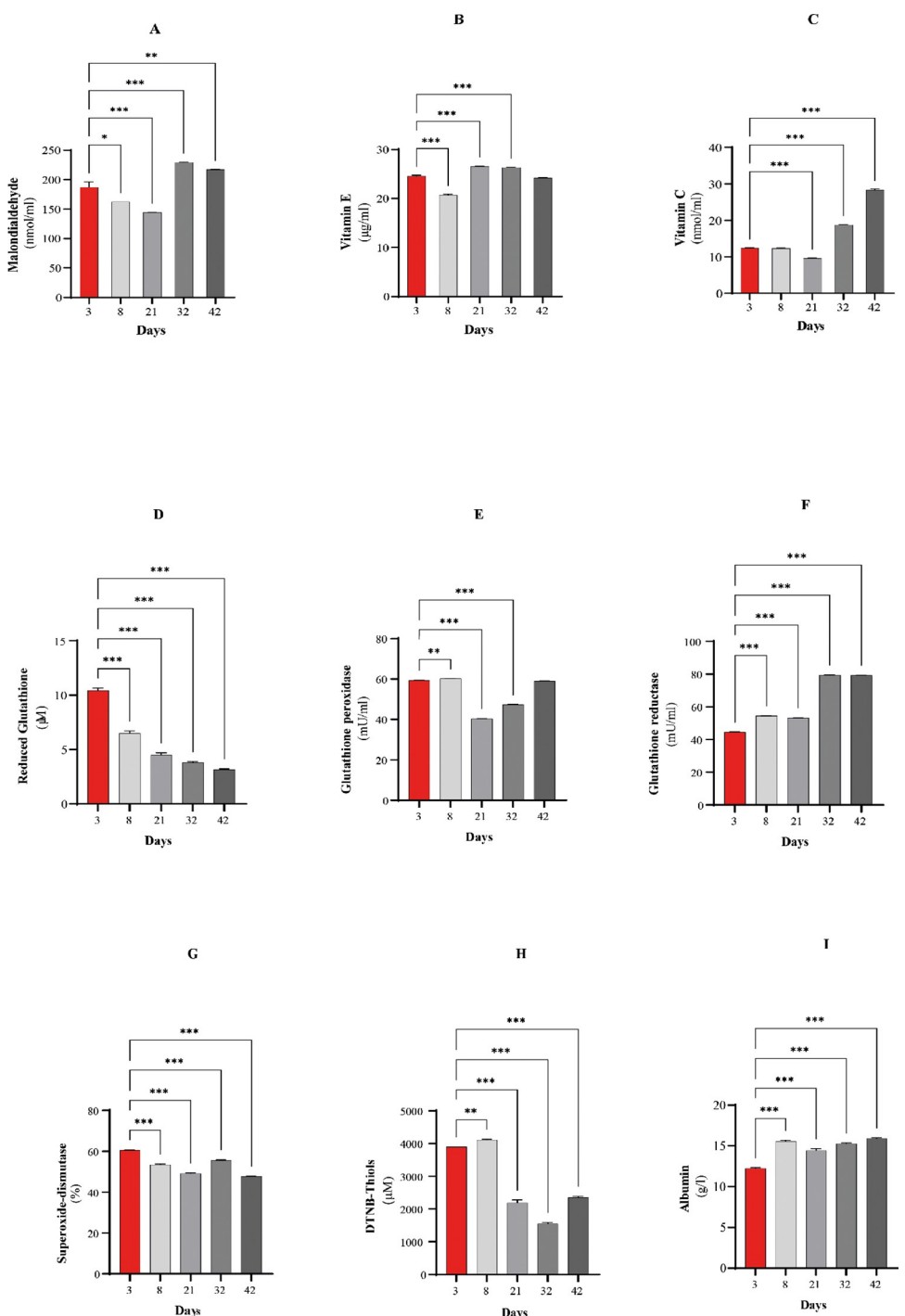

**Fig 1. Changes in the redox parameters in blood plasma at Days 3, 8, 21, 32, and 42.** (A) MDA concentration. (B) Vitamin E concentration. (C) Vitamin C concentration. (D) GSH concentration. (E) GPx activity. (F) GR activity. (G) SOD inhibition rate. (H) DTNB-thiol concentration. (I) Albumin concentration. Significant differences were determined by comparing the data from each time point with the data from the first time point. Data are expressed as the means ±8 SEMs; *P<0.05, ** p<0.005, *** p<0.001.

significantly until the end of the experimental period. In contrast, the blood plasma vitamin E concentration was the highest on Day 21 (26.6 ng/ml), but by the end of the experiment (Day 42), it decreased (Fig 1B). As the MDA concentration increased, the amount of α-tocopherol decreased. This is not surprising because if the contents of pro-oxidants, which cannot be eliminated by the redox system, increase, the hydroxyl radicals from the Fenton reaction can react with lipid molecules and give MDA [74–76]. Tocopherol behaves as a $H^+$ donor and transforms into α-tocopherol radicals after reaction with $OH^-$ radicals, which are regenerated by vitamin C [77–79].Therefore, we can conclude that the antioxidant status of the animals declined and the number of lipid peroxidation processes increased.

Similar to that of MDA, the blood plasma vitamin C concentration was the lowest on Day 21 and then increased significantly by the end of the experiment it increased significantly ($p < 0.000001$) (Fig 1C). As the amount of free radicals increases, the body is supposed to eliminate them, and in turn, the synthesis of ascorbic acid increases in the liver. The α-tocopherol radical is transformed into α-tocopherol and inhibits lipid peroxidation processes [80,81]. Thus, it can be concluded from our results that the elevation of the antioxidant level was not neutralized by the extensive synthesis of pro-oxidants.

The GSH concentration showed a continuous, significant ($p < 0.000001$) decrease throughout the experiment (Fig 1D). Hydrogen peroxide ($H_2O_2$) was eliminated by GPx using GSH. GSH is the most important small molecule antioxidant in the body, which eliminates pathological free radicals and lipid peroxides. GSH has a significant role in the redox cycle of cells [82,83].

In birds, this enzyme has an important role in the maintenance of redox homeostasis. In spite of the decreasing GSH level, by breaking down the large amount of glutathione disulfide (GSSG) produced, GR contributes to the maintenance of the activity of glutathione-dependent GPx. GPx activity significantly decreased by Day 21 ($p < 0.000001$) and then significantly increased until the end of the experiment ($p < 0.000001$) (Fig 1E).

The GSH level decreased although the activity of GR, which cleaves conjugated glutathione, showed a continuous significant increase ($p = 0.000036$) during the 42 days (Fig 1F).

GSH is structurally a tripeptide (Gly-Cys-Glu) [84] that protects enzymes with thiol groups, e.g., GR [EC 1.6.4.2] and GPx [EC 1.1.1.9], from inactivation. It is a cofactor of GPx [85,86] and a strong reducing agent that is oxidized during these reactions, while oxidized glutathione forms through the SH-bridge. In the reaction catalyzed by the NADPH-dependent GR, oxidized glutathione is converted back into GSH at a specific rate (the characteristic rate of the reduced to oxidized form in a healthy body is 500:1) [72,75,76]. The glutathione concentration decreased by almost half even with intensive GR in 21-day-old birds, indicating significant oxidative stress (Fig 1). This reduction in GPx activity had adverse effects on the regeneration and detoxification pathways, implying an increase in the vitamin E blood plasma concentration. However, lipid peroxidation was not compensated for, resulting in elevated MDA levels [87,88]. This raises the issues of when GSH diminishes and which biochemical processes are able to relieve the body of the abundant pro-oxidants.

The SOD [EC 1.15.1.1] inhibition rate significantly decreased ($p < 0.000001$) by Day 21 and slightly increased ($p < 0.000001$) by Day 32 (Fig 1G), and an additional decrease was observed at the end of the experimental period ($p < 0.000001$). The plasma free thiol concentration significantly decreased by Day 21 ($p < 0.00001$), and then a further decrease was observed on Day 32 (Fig 1H). A significant increase was detected in albumin concentration until Day 42 ($p < 0.000001$) (Fig 1I).

The decrease of antioxidants such as glutathione and increases of radicals (superoxide radical, hydroxyl radicals) leads to the increasing then decreasing inhibition rate of SOD [89,90]. Reduced thiols and low molecular-weight antioxidants are called second line of defense.

Oxidative stress, ROS generation cause oxidative modifications and decrease free thiol concentration [91,92].

According to the latest research, large molecule antioxidant proteins are responsible for the antioxidant status of plasma [93]. In the blood plasma of birds, albumin is found in the largest concentration (average 12–15 g/l) [94]. During development and growth, albumin levels are stable in plasma, and significant changes might lead to fatal consequences [95–97]. However, during certain periods, significant differences can be seen [94], and such changes can be explained by the rhythm of albumin exchange. The degree of freedom of the Cys binding site is responsible for the antioxidant properties of albumin [98–100].

Although the antioxidant capacity of albumin is generally not studied in animals, clinical data are already available regarding its role in the maintenance of redox homeostasis since it is the most abundant protein in the blood plasma. Medical use of human serum albumin (HSA) has been widely spread in critical cases (sepsis and cirrhosis) for a long time. HSA can not only maintain osmotic pressure but also has special antioxidant functions derived from its ability to scavenge free radicals [101]. Its principal ligands, intermediate ionic forms of copper ($Cu^{2+}$) and iron ($Fe^{2+}$), are implicated in $H_2O_2$ elimination (Haber-Boss and Fenton reactions); thus, has works as an inhibitor of lipid peroxidation [102,103]. Moreover, because of its free Cys34 structure, it can reversibly bind GSH and Hcy, forming mixed disulfides [104]. Approximately two-thirds of HSA is in its reduced form mercaptoalbumin, while one-third is in the mixed disulfide form, in which Cys can bind superoxide anions and be oxidized to the stable sulfenic acid form (HSA-SOH). Therefore, HAS has gained increasing attention in poultry [98]. Complex structural analysis of chicken serum albumin (CSA) has determined that Cys has the same function in CSA as in HSA. The position of Cys in proteins is highly conserved, and Cys can generally be found in functional or structural regions responsible for catalytic metal binding and/or redox regulating activity [105,106]. Our investigation of the saturation of CSA during the whole lifecycle revealed that the free binding site of CSA was saturated owing to oxidative stress on Day 21. CSA capacity analyses might provide data regarding redox homeostasis and the health status of animals.

## The concentration of thiols

The total plasma concentration of thiols and disulfides (TTD) is shown in Fig 2. At every time point, the participants of Cys homeostasis and TSP were examined.

During the TTD measurements, we focused on the participants in Cys homeostasis and TSP (Fig 2). The results from the thiol derivatives corresponded well with the alterations in antioxidant parameters. Changes in GSH concentration showed similar results as those from the colorimetric assay (Fig 2B). The most salient decrease in GSH levels was detected on Day 21 (p<0.000001), which slightly decreased further until the end of the experimental period. This was in contrast with the γ-GC concentration, which is the precursor of GSH [107]. The concentration of γ-GC increased by Day 8 and remained almost unchanged thereafter (Fig 2F).

Significant changes in the concentrations of GSH and CySS, the oxidized dimer form of Cys [108], corresponded well with the alterations in the disulfide Cys-Gly level (Fig 2B, 2F and 2G).Cys-Gly is formed during the degradation of GSH and CySS [109]. The amounts of GSH and CySS decreased after a solid initial rise, while that of Cys-Gly decreased at first and then increased.

Cys, one of the components of GSH, is known to be an important structural and functional element of proteins because of its free thiol group [110]. In the extracellular space, its free form is very reactive, which in turn can cause the generation of toxic free radicals. [26,111].

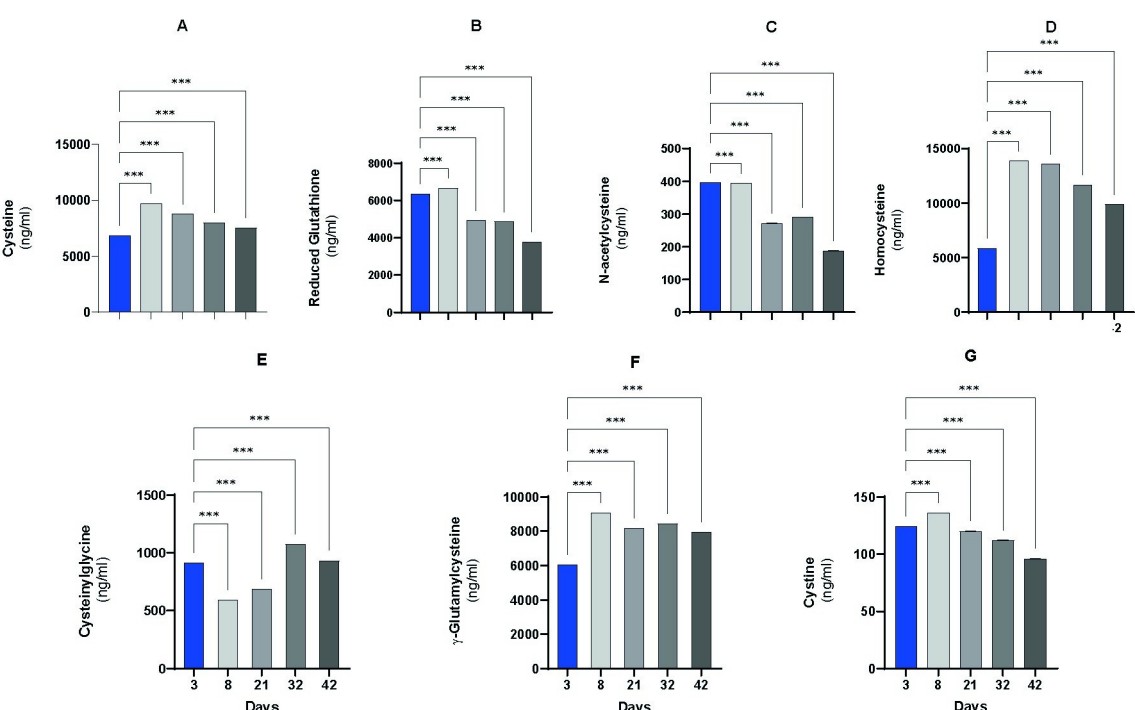

**Fig 2. The concentration of total thiols in the blood plasma on Days 3, 8, 21, 32, and 42.** (A) Cys concentration. (B) GSH concentration. (C) NAC concentration. (D) Hcy concentration. (E) Cys-Gly concentration. (F) γ-GC concentration. (G) Significant differences in CySS were detected when comparing the data from the first time point with the data from each subsequent time point. Data are expressed as the means ± SEMs; *P<0.05, ** p<0.005, *** p<0.001.

Considering that protein-bound Cys is an important indicator of antioxidant status, it can be characterized well by estimating the bound and total Cys levels. Moreover, the GSH content in plasma is generally low, and GSH eliminates free radicals only intracellularly; thus, GSH content cannot provide enough information about antioxidant status [112,113]. Our measurements showed a significant elevation in the Cys level in fast-growing broilers on Day 8, and the Cys level remained high during the whole lifecycle of the birds (Fig 2A).

Hcy is an analog of Cys, which is formed from methionine and can be transformed into Cys in the TSP or reverted to methionine after remethylation [109,114]. The Hcy level was elevated by more than two fold on Day 8 and remained high until Day 42 (Fig 2D).

The concentration of NAC changes significantly during certain growth periods, and a continuous decrease was observed compared to that on the 3$^{rd}$ day.

The NAC measurement results are not surprising either since albumin saturation is very high in 21-day-old animals. One of the tasks of NAC is to breakdown the mixed disulfides of human serum albumin (HSA) and thereby increase the degree of freedom at the Cys34 binding site [115,116].

The crucial roles of free and bound thiols has already been proven in human studies, but there is not yet a suitable method for their measurement. A novel method published by Xiaoyun Fu et al. [58] provides a reliable and reproducible procedure to analyze all types of thiol compounds.

Thiols fulfil a prominent role in the three-level antioxidant protection system: (1) the reducing agent GSH, which is detectable in the extracellular space in lower amounts; (2) the regulation of redox status; and (3) immunological functions [117]. GSH is generated from CySS/Cys by the rate-limiting enzymes γ-GC synthetase (γ-GCS) and glutathione synthetase (GSS). In

addition to being a precursor in GSH synthesis, cystine/Cys is a critical substrate of protein synthesis as well as an extracellular reducing agent. In addition, precursors of Hcy are produced via the TSP [109,118].

GSH degradation is catalyzed by γ-glutamyl transpeptidase (γ-GT) to generate a Cys-Gly and a glutamic acid (Glu) residue exclusively in the extracellular space of mammalian cells. Glu is transferred to another acceptor amino acid, Cys-Gly, which is degraded further by a dipeptidase [119] These processes demonstrate that GSH homeostasis is a combination of its intracellular synthesis and utilization as well as export and extracellular degradation. We were able to detect all of the abovementioned components in the blood plasma of fast-growing broilers (Fig 2) [120–123].

Thiols are present in the cells in millimolar concentrations mainly in their reduced forms. The total glutathione concentration is 2–17 mM, 91% of which is GSH [124,125]. In contrast, in the extracellular space and especially the plasma, thiols are rapidly oxidized, resulting in a lower concentration, 0.4–0.6 mM, in the blood plasma. The concentrations of small thiol molecules, such as Cys, Cys-Gly, GSH, Hcy and γ-GC, are12-20 μM, and the total GSH content is 50–55%. In addition, Hcy can form disulfide bridges and be produced from thiols and might modify lysine residues, which in turn can lead to N-homocysteine residues in peptides [26].

The measurement results suggest that there is a close connection between the defense system that maintains redox homeostasis and the TSP (Fig 3) [109] pool. In this system, the measured change in Hcy concentration can be of diagnostic importance, as it predicts the emerging GSH deficiency, the increase in the expression level of inflammatory cytokines, and

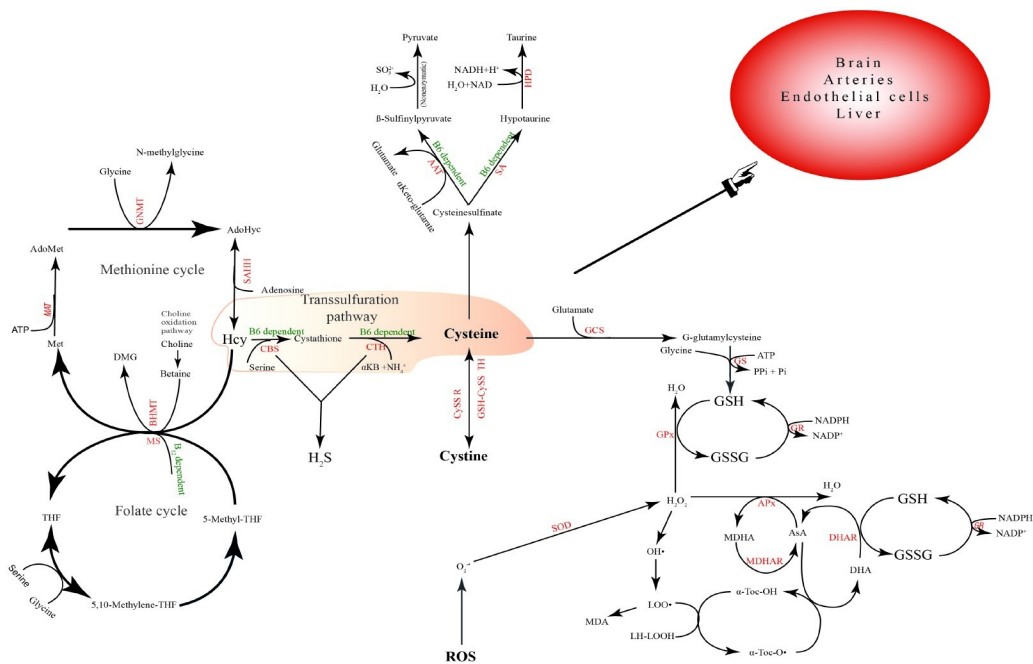

**Fig 3. Transsulfuration pathway and antioxidant system.** Tetrahydrofolate (THF); 5,10-methylene-tetrahydrofolate (5,10-Methylene-THF); methionine synthase (MS); methionine (Met);betaine homocysteine methyltransferase (BHMT); dimethylglycine (DMG); S-adenosylmethionine (SAM); methionine adenosyltransferase (MAT); S—adenosylmethionine (AdoMet); glycine N-methyltransferase (GNMT); S-adenosylhomocysteine (AdoHcy); S-adenosylhomocysteine hydrolase (SAHH); homocysteine (Hcy); cystathionine β-synthase (CBS); cystathionine γ-lyase (CTH); α-ketobutyrate (αKB); aspartate aminotransferase (AAT); sulfinoalanine decarboxylase (SAD); Hypotaurine dehydrogenase (HPD); cystine reductase (Cyss R); Glutathione-cystine transhydrogenase (GSH-Cyss TH); γ-glutamylcysteine synthase (GCS); glutathione synthase (GS); and aspartateaminotransferase(AST).

the development of nonalcoholic fatty liver disease [126,127]. The consequence of liver tissue damage is the increased concentration of necroenzymes in the blood [128,129]. Another consequence of damaged liver tissue is that it cannot express the enzymes cystathionine β-synthase (CBS) and cystathionine γ-lyase (CTH), which are responsible for breaking down Hcy [109]. This has further consequences, e.g., there is a lack of $H_2S$, which results in the redox balance being disturbed, which generates new inflammatory processes through the NF-κB pathway [130,131].

## Relationship between the antioxidant system and inflammatory markers

Cytokine levels increased in parallel with the weakening antioxidant capacity as the birds grew (Fig 4). We observed an approximately fourfold increase in all parameters from Day 21.

IL-2, IL-6, IL-8 and TNF-α have crucial roles in stimulating, proliferating and differentiating T-helper cells, B cells and natural killer cells as well as in the recruitment of other immune cells [132–134]. Our results also showed that the concentrations of these markers were extremely high on Day 42, which denotes severe inflammation in the animals. Oxidative stress affects the immune function of chickens. Cytokines have a major immunomodulatory role. The increased production of ROS resulting the weakening of the first and second lines of defense, results the increased concentration of inflammatory cytokines in the plasma [135]. However, the most appropriate marker and time point for measurement to evaluate inflammatory status as accurately as possible is still elusive.

The transcription factor nuclear factor κB (NF-κB) is involved in inflammation, the stress response, and cell differentiation, proliferation and apoptosis. Moreover, it regulates the gene expression of pro-inflammatory cytokines, chemokines, adhesion molecules, receptors and microRNA [136,137]. Oxidative stress and intracellular redox status have been proven to be essential in NF-κB activation; $H_2O_2$ can enhance, whereas antioxidants block, this process [138].

The extracellular Cys/CySS redox potential in blood plasma exposed to oxidative stress has been shown to evoke monocyte adhesion to vascular endothelial cells, activate NF-κB and enhance inflammatory cytokine expression [139,140]. In particular, interleukin 1β (IL-1β) and tumor necrosis factor alpha (TNF-α) induce the expression of their own and other cytokines. The identification of the primary disorder needs to be mentioned here. Oxidative stress that

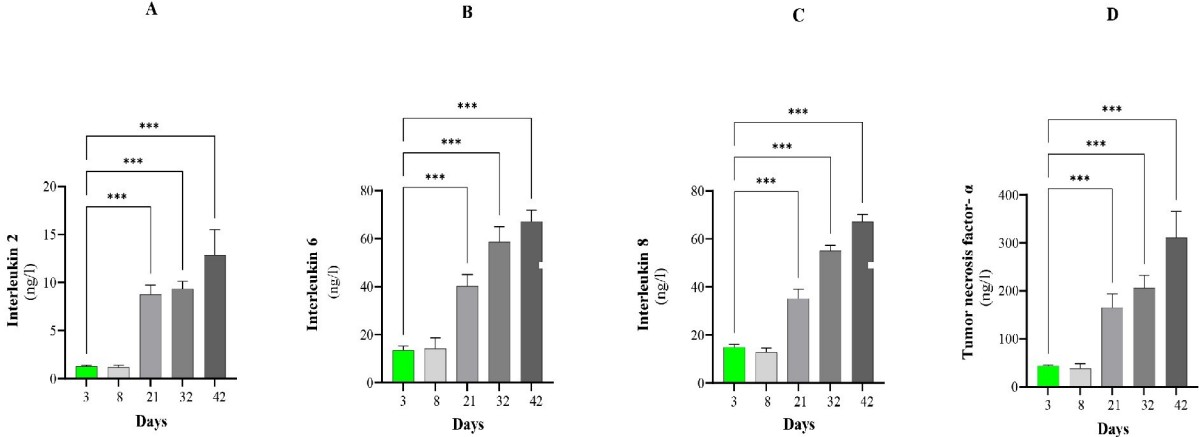

**Fig 4. The concentrations of inflammatory markers in the plasma at Days 3, 8, 21, 32, and 42.** (A)IL-2 concentration. (B) IL-4concentration. (C) IL-6concentration. (D) TNF-α concentration. Significant differences were determined by comparing the data from each time point with the data from the first time point. Data are expressed as the means ± SEMs; *P<0.05, ** p<0.005, *** p<0.001.

develops first in one organ leads to inflammation and enhances oxidative stress. In contrast, if inflammation is the primary disorder, it promotes oxidative stress and further increases inflammatory processes. Our data demonstrated upregulated proinflammatory cytokine expression in birds on the 21st day, which indicates the evolution of strong inflammation [141].

Reactive oxygen species (ROS) target thiol compounds of specific proteins that participate in signaling pathways implicated in cell proliferation, differentiation and apoptosis (kinases, phosphatases, and transcription factors). Moreover, hepatocytes are damaged, and ROS indicate their apoptosis, which is known to be important, especially in nonalcoholic steatohepatitis (NASH), where ROS are created in multiple ways [142–144].

ROS production and mitochondrial dysfunction are a result of complex metabolic processes, including enhanced β-oxidation and inhibition of the mitochondrial electron transport chain by TNF-α and lipid peroxidation products. Mitochondrial damage results in the secondary inhibition of β lipids, and oxidation increases steatosis levels [145].

Oxidative stress and antioxidant protection have also been detected in fibrosis and cirrhosis in clinical and animal studies. DNA lipids and proteins disintegrated by oxidative stress lead to necrosis, the loss of hepatocytes and an escalating inflammatory response, which in turn can cause fibrosis [146]. Inflammation evoked by necrosis in the priming phase of hepatic impairment contributes to removing cellular debris and promoting liver regeneration, ensuring restoration of the liver structure and function following acute damage [146,147]. Progressive liver fibrosis and chronic inflammation can develop in cases when the primary disease and sustained stimuli are not controlled, which together with necrosis, are characteristic of fibrotic liver disorders [148]. This was confirmed by our results from 21-day-old birds, namely, the level of AST, a necrotic enzyme, was increased significantly.

The plasma aspartate aminotransferase (AST) concentration significantly increased on Day 21($p<0.000001$), then on both Day 32 ($p = 0.000009$) and Day 42 ($p = 0.000003$) compared to the initial day (Fig 5). After Day21, it moderately decreased but remained very high until Day 42.

Adaptive immune responses provoked by oxidative stress might be independent predictors of fibrotic alterations in nonalcoholic fatty liver disease (NAFLD) [145]. The pathogenesis of NAFLD is extremely complex and accompanied by modified molecular biological and metabolic pathways. The TPS is involved in these events and is correlated with steatosis, insulin resistance, oxidative stress, ER stress, inflammation and portal hypertension [109]. The correlation between the TSP and oxidative stress has already been proven since TSP is a regulator of GSH production. Elevated plasma Hcy levels, steatosis, oxidative stress and fibrosis were found in mice with cystathionine beta synthase (CBS) deficiency [149,150]. The expression of genes implicated in ER stress, liver lipid homeostasis and steatosis seems to be upregulated by CBS deficiency, while knockout of cystathionine gamma synthase (CSE) causes a decrease in liver lipolysis [151–153].

TSP coverts Hcy into Cys via the intermediate cystathionine and has a key role in sulfur metabolism and the cellular redox system, which is considered to be the only pathway of Cys biosynthesis. The first step is catalyzed by vitamin B6-dependent CBS to generate cystathionine via a condensation reaction using Hcy and serine as substrates. Cystathionine is then hydrolyzed by vitamin B6-dependent CSE to form Cys and α-ketobutyrate (aKB) [109].

It is important to note that Hcy-Cys conversion is irreversible. Therefore, all downstream pathways of TSP are part of Cys metabolism. The nonessential amino acid Cys is a crucial participant in sulfur metabolism and a precursor of significant metabolites, including GSH, $H_2S$, sulfate and taurine. A lack of these molecules has been proven to generate oxidative stress and ultimately induce inflammation. Damage to hepatocytes downregulates the expression of the

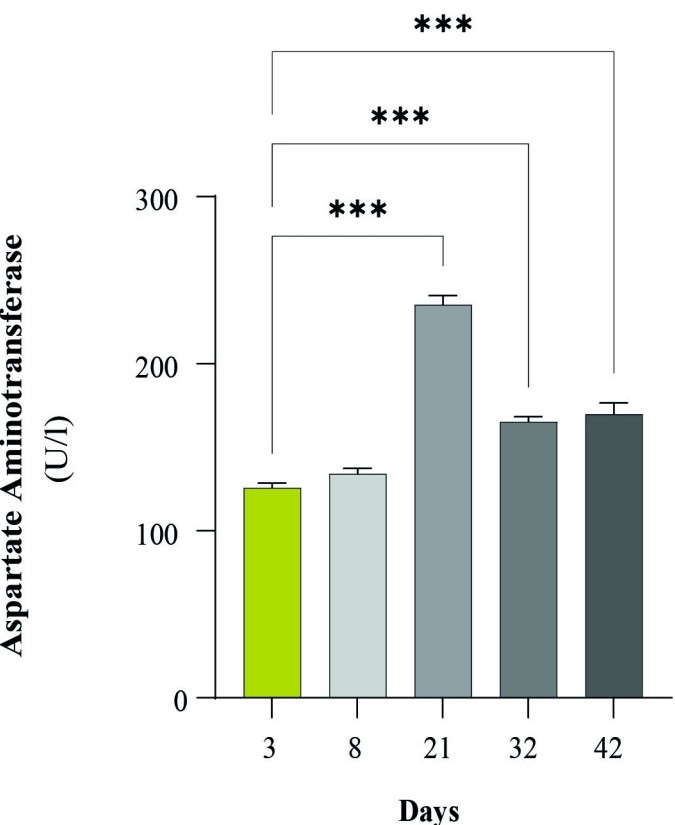

**Fig 5. The concentration of AST in plasma at Days 3, 8, 21, 32, and 42.** Significant differences were detected when comparing the data from the first time point with the data from subsequent time points. Data are expressed as the means ± SEMs; *$P<0.05$, ** $p<0.005$, *** $p<0.001$.

Hcy-degrading enzymes CBS and CSE, and the accumulation of Hcy hampers the glutathione pool, which finally creates a vicious cycle [109,118,154].

## Correlation between thiol compounds, cytokines and antioxidant parameters

The enzymes responsible for Hcy degradation outside the liver are expressed in endothelial cells in the wall of the aorta and in the brain. The resulting systemic diseases are likely to be related to elevated Hcy concentrations.

The relationships between the measured antioxidant parameters were determined by correlation analysis. The correlations are illustrated on a heatmap with a gradient of colors, where the red scale represents negative correlations and the blue scale represents positive correlations (Fig 6). The concentrations of total cysteinylglycine, IL-6, IL-8 and TNFα increased in parallel with the increase in MDA concentration. The increased activity of GR is negatively correlated with the concentration of CySS. This shows that the consequence of the increase in lipid peroxidation processes is the decomposition of GSH, since the concentration of the decomposition product increases but the amount of the precursor, which is indispensable for its synthesis, decreases. This helps with the initiation of inflammatory processes [155–157]. Presumably, the body tries to protect itself against damage to the membrane system by synthesizing vitamin C

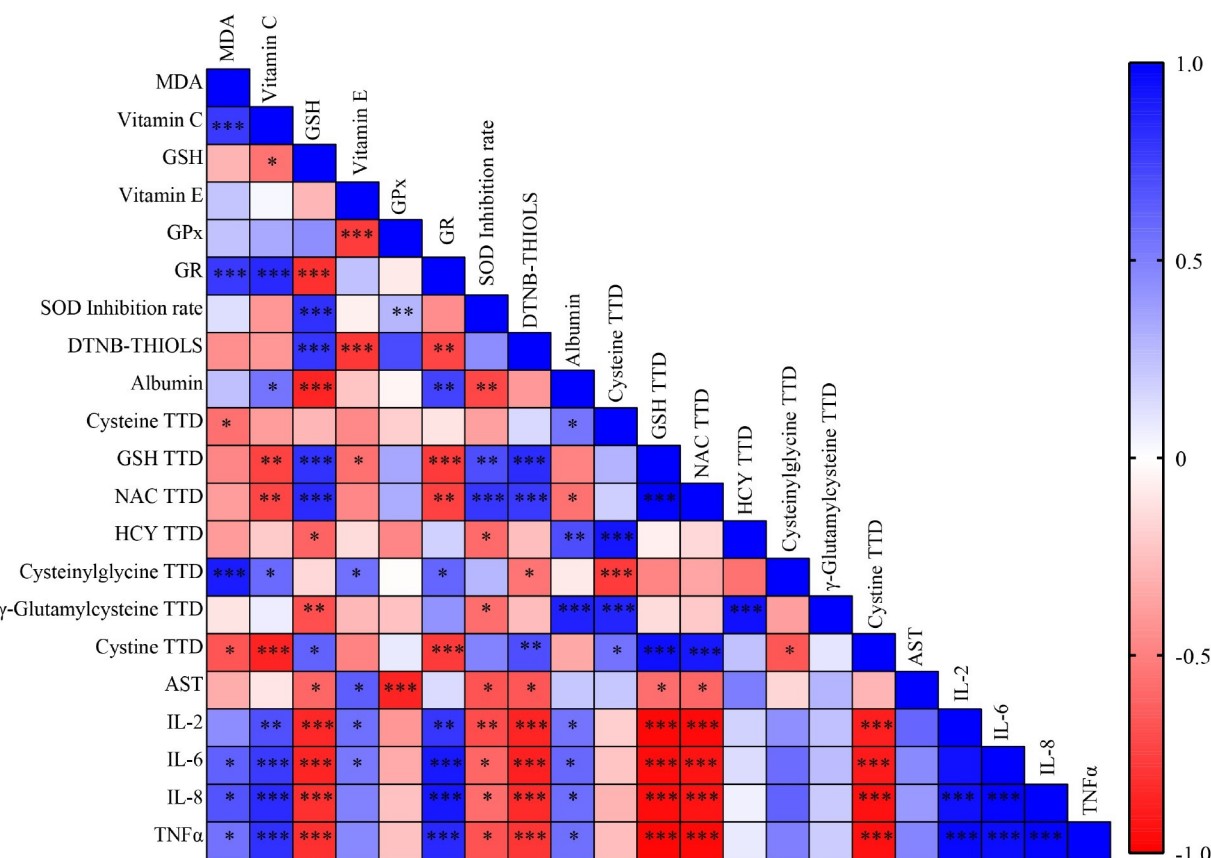

**Fig 6. Correlations between antioxidant parameters, cytokines and thiols.** *, **, *** indicate significant differences at P < 0.05, P < 0.01, and P < 0.001, respectively.

[158]. In the case of GSH, we observed correlations that were contrary to this since the activity of GR decreased in the presence of GSH. Moreover, since a suitable reduced form is available in the defense system, SOD activity is high, albumin saturation is low, and there are many forms of free thiols in plasma. GSH levels are also negatively correlated with the level of inflammation. If albumin saturation in the plasma is low, then the concentrations of thiol components, including GSH, is high, so the redox system is not damaged and inflammatory cytokines are expressed to a lesser extent [106].

## Histological alterations to the liver

In addition to increasing levels of the main inflammatory indicators, structural alterations to the liver were clearly seen in histological sections (Fig 7), and included hydropic degradation, the infiltration of inflammatory cells, and serious intrahepatic bleeding [17].

According to the histological image of the liver (Fig 7), the number of lipid deposits characteristic of fatty degeneration of the liver also increased significantly in the 42-day animals. This was also reflected in the number of Mallory bodies, indicating the aggregation of damaged intermediate filaments [159,160].

## Conclusion

In our study, growth performance did not differ from the typical literature values for fast-growing broilers. It can be seen that 21 days is critical time point for birds, and a large increase

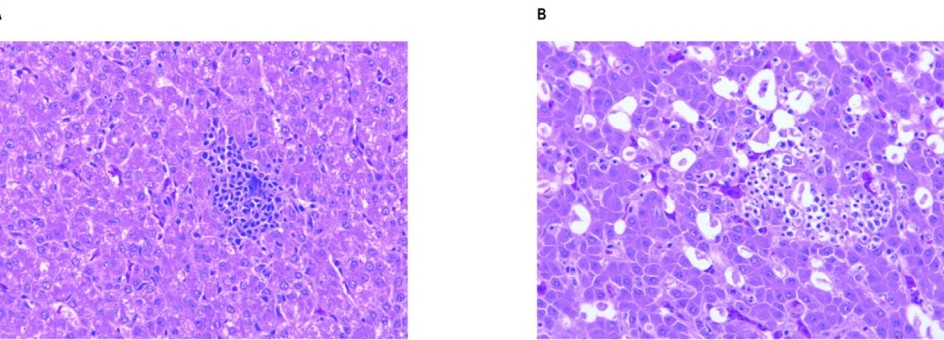

**Fig 7. Representative micrographs of a chicken liver.** (A) Section taken at Day21. (B) Section taken at Day 42.

in body mass is associated with the formation of many free radicals. The antioxidant defense system cannot neutralize the intense pro-oxidant synthesis. There is a close connection between redox homeostasis and the TSP. According to the Hcy concentration and histological alterations, nonalcoholic fatty liver disease may developat42 days. This was evidenced by the increased lipid deposition that is characteristic of fatty degeneration of the liver. Monitoring changes in Hcy concentration and find critical growth stage could be of diagnostic importance. Necessary to look for new biomarkers to measure that are easily recognized when the animals' redox balance is disturbed because it cannot be detected visually. From an animal welfare point of view, this effect is not negligible.

## Acknowledgments

We gratefully thank Gábor Méhes director of Pathology University of Debrecen, for the histological examinations. We are grateful to all the coworkers of the Kismacs Experimental Station of Animal Husbandry.

## Author Contributions

**Conceptualization:** László Stündl, Judit Remenyik.

**Data curation:** Georgina Pesti-Asbóth, Piroska Bíróné Molnár, Melinda Paholcsek.

**Funding acquisition:** Judit Remenyik.

**Investigation:** Georgina Pesti-Asbóth, Endre Szilágyi, Piroska Bíróné Molnár, Zoltán Cziáky.

**Methodology:** Georgina Pesti-Asbóth, János Oláh, László Babinszky, Levente Czeglédi, Zoltán Cziáky, Judit Remenyik.

**Project administration:** Georgina Pesti-Asbóth.

**Resources:** Judit Remenyik.

**Supervision:** Endre Szilágyi, János Oláh, Judit Remenyik.

**Visualization:** Georgina Pesti-Asbóth, Levente Czeglédi, Melinda Paholcsek, László Stündl.

**Writing – original draft:** Georgina Pesti-Asbóth, László Babinszky, Judit Remenyik.

**Writing – review & editing:** Georgina Pesti-Asbóth, László Babinszky, Judit Remenyik.

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
