## [Decision Letter · Decision Letter 0]

4 Jul 2023

PONE-D-23-15244Monitoring physiological processes of fast-growing broilers during the whole life cycle: changes of redox-homeostasis effected to trassulfuration pathway predicting the development of non-alcoholic fatty liver diseasePLOS ONE

Dear Dr. Remenyik,

Thank you for submitting your manuscript to PLOS ONE. After careful consideration, we feel that it has merit but does not fully meet PLOS ONE’s publication criteria as it currently stands. Therefore, we invite you to submit a revised version of the manuscript that addresses the points raised during the review process.

We look forward to receiving your revised manuscript.

Kind regards,

Mohammed Fouad El Basuini, Professor

Academic Editor

PLOS ONE

Journal Requirements:

"This study was financially supported by the Gazdaságfejlesztési és Innovációs Operatív Program (GINOP) GINOP-2.3.2-15-2016-00042 project of the Széchenyi 2020 Program given by the European Union and the Hungarian Government" 

Reviewers' comments:

Reviewer's Responses to Questions

**Comments to the Author**

1. Is the manuscript technically sound, and do the data support the conclusions?

Reviewer #1: Yes

Reviewer #2: Yes

2. Has the statistical analysis been performed appropriately and rigorously? 

Reviewer #1: No

Reviewer #2: Yes

3. Have the authors made all data underlying the findings in their manuscript fully available?

Reviewer #1: Yes

Reviewer #2: Yes

4. Is the manuscript presented in an intelligible fashion and written in standard English?

Reviewer #1: Yes

Reviewer #2: Yes

5. Review Comments to the Author

Reviewer #1: Manuscript is well conceptualized and written well, however, some minor changes required specially statistical analysis section and results section. I have mentioned all my suggestions/corrections in attached file.

Reviewer #2: The manuscript may be accepted with some minor corrections. The research is found well. FCR and Body gain are also found sound and that is very important in the broiler industry. With regard to the food safety, the research will be helpful for the policy level, professionals and academician.

6. PLOS authors have the option to publish the peer review history of their article (what does this mean?). If published, this will include your full peer review and any attached files.

Reviewer #1: No

Reviewer #2: No

While revising your submission, please upload your figure files to the Preflight Analysis and Conversion Engine (PACE) digital diagnostic tool, https://pacev2.apexcovantage.com/. PACE helps ensure that figures meet PLOS requirements. To use PACE, you must first register as a user. Registration is free. Then, login and navigate to the UPLOAD tab, where you will find detailed instructions on how to use the tool. If you encounter any issues or have any questions when using PACE, please email PLOS at figures@plos.org. Please note that Supporting Information files do not need this step.<quillbot-extension-portal></quillbot-extension-portal>

---

## [Author Response · Author response to Decision Letter 0]

3 Aug 2023

Reviewer #1

“How sample size was determined?”

The pen size in the experimental barn determined the number of birds to achieve a stocking density used in broiler chicken production (3.9 m2 pens, 60 birds = stocking density of 650 cm2/bird)

“What was experimental design?” and “Which experimental design?”

Birds were considered as biological replicates for the measured parameters except feed intake, where the three pens provided three data.

“Mention strain”

We corrected it. Page no.6, line no.114 “ Ross 308 broiler chickens”.

“Why too old reference was used when latest Strain Based guidelines are available.”

Corrected. Page no.6, line no.121.,

In References:

Page no.35., line no. 769-770

30. Ross-BroilerHandbook2018-EN.pdf. Available: https://aviagen.com/assets/Tech_Center/Ross_Broiler/Ross-BroilerHandbook2018-EN.pdf

 “Add references of each parameter”

We filled the missing citation.

Page no. 9, line no 168 [33]

In references: page no.35 line no. 778-781 

33. Santolim LV, Amaral MEC do, Fachi JL, Mendes MF, Oliveira CA de. Vitamin E and caloric restriction promote hepatic homeostasis through expression of connexin 26, N-cad, E-cad and cholesterol metabolism genes. The Journal of Nutritional Biochemistry. 2017;39: 86–92. doi:10.1016/j.jnutbio.2016.09.011

Page no. 13, line no. 256. citation [57]

In references: page no.37, line no.855-857

57. Gane EJ, Weilert F, Orr DW, Keogh GF, Gibson M, Lockhart MM, et al. The mitochondria-targeted anti-oxidant mitoquinone decreases liver damage in a phase II study of hepatitis C patients. Liver International. 2010;30: 1019–1026. doi:10.1111/j.1478-3231.2010.02250.x

Page no. 15, line no.302.; citation [58]

In references: page no. 37, line no.858-860

58. Fu X, Cate SA, Dominguez M, Osborn W, Özpolat T, Konkle BA, et al. Cysteine Disulfides (Cys-ss-X) as Sensitive Plasma Biomarkers of Oxidative Stress. Scientific Reports. 2019;9: 115. doi:10.1038/s41598-018-35566-2

Page no. 16, line no. 325 citation [59-62]

In reference: page no. 37., line no. 861-875

59. Wandita TG, Joshi N, Nam IS, Yang SH, Park HS, Hwang SG. Dietary Supplementation of Purified Amino Acid Derived from Animal Blood on Immune Response and Growth Performance of Broiler Chicken. Tropical Animal Science Journal. 2018;41: 108–113. doi:10.5398/tasj.2018.41.2.108

60. Varasteh S, Braber S, Akbari P, Garssen J, Fink-Gremmels J. Differences in Susceptibility to Heat Stress along the Chicken Intestine and the Protective Effects of Galacto-Oligosaccharides. PLOS ONE. 2015;10: e0138975. doi:10.1371/journal.pone.0138975

61. Krzysica P, Verhoog L, de Vries S, Smits C, Savelkoul HFJ, Tijhaar E. Optimization of Capture ELISAs for Chicken Cytokines Using Commercially Available Antibodies. Animals. 2022;12: 3040. doi:10.3390/ani12213040

62. Jarosz ŁS, Marek A, Grądzki Z, Kwiecień M, Kaczmarek B. The effect of feed supplementation with a copper-glycine chelate and copper sulphate on selected humoral and cell-mediated immune parameters, plasma superoxide dismutase activity, ceruloplasmin and cytokine concentration in broiler chickens. J Anim Physiol Anim Nutr. 2018;102: e326–e336. doi:10.1111/jpn.12750

„add mathematical model for better illustration of data analysis”

Corrected, page no.16, line no. 331-332.

„The normality assumption was checked by analyzing the ANOVA model residuals using Shapiro-Wilk test. The homogeneity of variance was checked using Levente’s test.” 

„add actual p-value of each result” and„add SEM and actual p-value”

Corrected, page no. 17, line no. 343.; 345.; 347-348, 356., Table 2.

„add logical reasoning of each result”

We revised the section „ Results and Discussion” and corrected.

Page no.22, line no. 434-438.

Page no.26-27, line no.557-560

„should be brief and to the point”

We revised „ Conclusion” and corrected.

Reviewer #2

“Page no. 6, line no. 102: Separate “abovementioned”; it is jumbled.”

Corrected.

“Page no. 6, line no. 114: “Day old” means “first day of chick’s life”. Please avoid to mention “1-day old” from the entire manuscript.”

Corrected in the entire manuscript

“Page no. 7, line no. 134: Replace “1” with “single”. Replace “3 birds/pen” with “three birds/ pen”.

Corrected.

“Page no. 9, line no. 157: Replace “dH2O” with “distilled water” and make it uniform for the entire manuscript.”

Corrected in the entire manuscript.

“Page no. 13, line no. 256: Insert “.” before the word “Both”.

Corrected.

“Abstract section: “A method was developed to measure the levels of other small molecule thiol components of plasma”.

But in page 12, line 234: The free thiol concentration was determined through Ellman’s assay (Ethos Biosciences Inc., Philadelphia, PA, United States).Please justify the statement of abstract?”

The free thiol concentration was determined through Ellman’s assay. We also determined the total thiol and disulfide content with a new method based on. concentration (protein bound and free) Xiaoyun Fu et al.’s publication

“Page 13, line 260 – 264: If any new method cited, it should be described by the authors. In case already published method, reference could be cited and a brief account could be presented.”

Corrected. Page no. 13-14, line no. 263-271

“Avoid presentation of conclusion in bullet form. It should be presented in sentence case within word count of 150.”

Corrected

---

## [Editor Report · Decision Letter 1]

7 Aug 2023

Monitoring physiological processes of fast-growing broilers during the whole life cycle: changes of redox-homeostasis effected to trassulfuration pathway predicting the development of non-alcoholic fatty liver disease

PONE-D-23-15244R1

Dear Dr. Remenyik,

We’re pleased to inform you that your manuscript has been judged scientifically suitable for publication and will be formally accepted for publication once it meets all outstanding technical requirements.

Kind regards,

Mohammed Fouad El Basuini, Professor

Academic Editor

PLOS ONE

Additional Editor Comments (optional):

Congratulations
---

## [Editor Report · Acceptance letter]

10 Aug 2023

PONE-D-23-15244R1 

Monitoring physiological processes of fast-growing broilers during the whole life cycle: changes of redox-homeostasis effected to trassulfuration pathway predicting the development of non-alcoholic fatty liver disease 

Dear Dr. Remenyik:

I'm pleased to inform you that your manuscript has been deemed suitable for publication in PLOS ONE. Congratulations! Your manuscript is now with our production department. 

Kind regards, 

on behalf of

Dr Mohammed Fouad El Basuini 

Academic Editor

PLOS ONE